# A New Proposal: A Digital Flow for the Construction of a Haas-Inspired Rapid Maxillary Expander (HIRME)

**DOI:** 10.3390/ma13132898

**Published:** 2020-06-28

**Authors:** Mauro Cozzani, Salima Antonini, Daniela Lupini, Davide Decesari, Fabrizio Anelli, Tiziana Doldo

**Affiliations:** 1Istituto Giuseppe Cozzani, 19125 La Spezia, Italy; maurocozzani@gmail.com; 2Department of Medical, Surgical and Health Sciences, School of Dentistry, Università di Trieste, 34100 Trieste, Italy; lupiniortodonzia@gmail.com; 3Private Lab, 24100 Bergamo, Italy; davidedecesari@gmail.com; 4Teor laboratorio specializzato in ortodonzia, 80 47923 Rimini, Italy; info@teorsrl.it; 5Dipartimento di Biotecnologie Mediche, Università di Siena, 53100 Siena, Italy; tiziana.doldo@unisi.it

**Keywords:** Haas-inspired rapid maxillary expander (HIRME), mixed dentition, rapid palatal expansion, digital flow, optical impression, stl file, 3D printer

## Abstract

Maxillary expansion is a common orthodontic treatment used for the correction of posterior crossbite resulting from reduced maxillary width. Transverse maxillomandibular discrepancies are a major cause of several malocclusions and may be corrected in different manners; in particular, the rapid maxillary expansion (RME) performed in the early mixed dentition has now become a routine procedure in orthodontic practice. The aim of this study is to propose a procedure that reduces the patient cooperation as well as the lab work required in preparing a customized Haas-inspired rapid maxillary expander (HIRME) that can be anchored to deciduous teeth and can be utilized in mixed dentition with tubes on the molars and hooks and brackets on the canines. This article thus presents an expander that is completely digitally developed, from the first moment of taking the impression with an optical scanner to the final solidification phase by the use of a 3D printer. This digital flow takes place in a CAD environment and it starts with the creation of the appliance on the optical impression; this design is then exported as an stl extension and is sent to the print service to obtain a solid model of the device through a laser sintering process. This “rough” device goes through a post-processing procedure; finally, a commercial expansion screw is laser-welded. This expander has all the advantages of a cast-metal Haas-type RME that rests on deciduous teeth; moreover, it has the characteristic of being developed with a completely digitized and individualized process, for the mouth of the young patient, as well as being made completely of cobalt-chrome, thus ensuring greater adaptability and stability in the patient’s mouth.

## 1. Introduction

Maxillary expansion is a common orthodontic treatment used for the correction of the posterior crossbite that results from reduced maxillary width; several treatment modalities are employed with similar objectives [1].

Transverse maxillomandibular discrepancies are a major cause of several malocclusions [2]. This type of malocclusion may be corrected in different manners, including slow expansion [3,4,5], rapid expansion [6,7,8], surgically assisted expansion [9,10] and miniscrew-assisted rapid palatal expansion (MARPE) [11,12,13].

As reported in the 2015 article by Mutinelli et al. [14], the rapid expansion of the maxilla is widely used to treat transverse deficiency [15], which means the opening of the mid-palatal suture [16] that has not yet completely ossified in growing individuals [17] and is characterized by a widening of the midpalatal suture, produced by forcing a lateral shift of the two horizontal processes of the maxilla.

Two different devices are used to achieve this type of expansion: the tooth-borne (Hyrax type) and the tooth-tissue-borne appliance (Haas type).

The main difference between the two devices is that the latter has an acrylic pad as reinforcement to the anchorage, and it covers the palatal mucosa bilaterally, where the activation screw is inserted in the center. In fact, the Hass expander is a tissue-borne fixed appliance anchored to the first bicuspids and molars by means of four abutment bands and to the palatal vault by acrylic masses. A variant of the traditional design is one in which the device is completely anchored to the deciduous teeth; in particular, to the deciduous second molars and to the deciduous canines.

Many authors have noted the effectiveness of this choice and analyzed the different positive aspects, and it has been seen that the main reason why deciduous teeth support this choice is to reduce the risk of the negative side-effects on permanent teeth produced by the expansion force [18] and/or by plaque accumulation around the bands: root resorption [16,19,20], bone loss [21,22], gingival recession [23] and white-spot lesions [24] on permanent dentition [14]. Furthermore, as demonstrated in the study conducted by Mutinelli et al. in 2015, the Haas expander anchored to the deciduous teeth is effective in increasing the width of the dental arch in patients with lateral crossbite; moreover, the index of anterior irregularity was lower among the patients undergoing expansion treatments compared to all the untreated study participants [25].

The misalignment in the anterior maxillary dentition improves so spontaneously, towards a low level of irregularity, that it can reduce the prevalence of the patient’s requests to align the anterior maxillary dentition. In particular, it has been seen that when a fixed orthodontic treatment is performed, the better the pre-treatment alignment of the permanent dentition, the better the long-term stability [14].

It seems that greater long-term stability is determined by the fact that by placing the expander on the deciduous molars, the permanent molar would move by movement of the basal bone only. This would ensure a greater stability than that which is achieved when the expander has its bands directly on the permanent molars, which, in this case, would move both through the movement of the basal bone and for the alveolar one (i.e., of their root inside the bone, with a buccal tipping of these teeth that partially relapses in the post-retention phase) [26].

Finally, another aspect that has been studied is that the deciduous teeth have a different anatomy to the permanent ones; in fact, the “bombé” is less accentuated, thus it is more difficult to find retention for the bands. Furthermore, the management of the young patient on the chair when we have to pick up the two bands—which is not easy, as we have highlighted above—and to take a transfer impression can be a difficult procedure, as well as requiring a work time of around 30/45 min to perform it [27].

It is precisely for these reasons that a new type of expander has been devised: a modified Haas-type Rapid Maxillary Expander (RME) appliance composed by a six-band metal-cast structure with a partial occlusal covering that bonds to the primary teeth using glass-ionomer cement. Clinicians see advantages in terms of the speed of application and patient compliance by taking a single impression [28]. The stability and retention of the appliance is guaranteed by the individual preparation of the castings and increased by the sandblasting of the parts in contact with the teeth as well as from the use of glass-ionomer cement [27].

Despite improvements in the process, the discomfort of taking the dental impression remains.

For this reason, dentistry has turned its interest toward the development of an optical detection system of the dental arches that allows the construction of impressions through three-dimensional CAD programs and the reconstruction of the desired devices through special three-dimensional printers.

This article describes an expander that has been developed with the innovative method described above.

The aim of this study is to propose a procedure that reduces patient cooperation as well as lab work in preparing a customized Haas-inspired rapid maxillary expander (HIRME) that can be anchored to the deciduous teeth and can be utilized in mixed dentition with a tube on the molars and hooks and brackets on the canines.

## 2. Procedure

What we present is a sintered metal device (DMLS/SLM), previously designed on the dental arch (in STL format), resulting from the intra-oral optical scan of the patient. In particular, it is a HIRME which is completely digitally designed and built, unlike the “classic” one which is built in the laboratory.

The main characteristic is that all this digital flow—up to the solidification of the device—takes place in a CAD environment and it starts with the creation of an STL file of the optical impression in order to obtain a solid model of the device, on which the expander is built (Figure 1a,b). The device rests completely on the deciduous teeth; in particular, on the second and first deciduous molars and on the deciduous canines. Moreover, if the use of elastics or a Delaire facemask is planned, hooks at the deciduous canines and/or 1.2 mm-diameter buccal tubes at the deciduous second molars are inserted. Once the drawing of the device is complete, the native file is exported in an STL extension and, once “validated”, sent to the print “service”. The product then returns to the laboratory only after the sintering of the metal (Figure 2a–e).

We also proceed to prototype the model of the dental arch in acrylic material with a special 3D printer (DWS D 20, 3DRPD, Mouilleron-le-Captif, France), which is used to position the “rough” device and carry out the subsequent phases, defined as the “post-printing process”, in the dental laboratory.

Once the “rough” structure is obtained, there will be a phase defined as the “post-printing process”: this operating phase consists in finishing the entire structure inside the HIRME with metal cutters and checking the undercut areas and the parts where the sintering has left roughness. Rubber pads are then used to finish the structure in all the margins that surround the teeth, in the frame of the arms and in the screw housing; finally, the entire external part of the HIRME is smoothed and polished. The screw (Forestadent—Palatal split screw type “S” 169-1323) is then positioned in the site already established on the drawing by welding it with the laser method to the structure, thus unifying the two metals and making them a unique part.

The finished product is placed on the model and transferred to the dentist (Figure 3a–c).

The final device is a Haas-inspired rapid maxillary expander that is constructed completely from cobalt-chrome, and it is customized to the patient’s mouth.

The dentist has only to fill the inner surface of the splints with a small quantity of glass-ionomer cement and then bond the appliance in place (Figure 4a–d).

## 3. Discussion

A previously described Haas-type RME appliance for primary anchorage was composed of two bands cemented on the primary second molars, two anterior lingual wires connected to the primary canines and an acrylic resin palatal pad with a central screw. The modified Haas-type RME consists of custom-made metal castings (one on each side) covering the first and second molars and the primary canines [28]. This type of device offers clinical advantages as it only requires an impression and no adaptation to the band, thus saving considerable session time and making the delivery appointment of the device much easier for young patients [27].

The assembly of a completely digitally developed and produced expander also overcomes the problem of taking a single impression too, since, in this case, it is carried out with a digital scanner. When the device comes back from the lab, it will already be perfectly adapted to the patient’s primary teeth, and a turbine-driven bur will not be required to create grooves for housing the anterior wires on the canines. The device is thus ready to be bonded in place by filling the inner surfaces of the splints with glass-ionomer cement. This makes the process much faster and easier than the one traditionally adopted in daily clinical practice.

Stability and retention are improved by the close adaptation of the bands to the coronal anatomy of the deciduous teeth, as well as the fact that, in this case, glass-ionomer cement is used for bonding. Finally, the increased stability is also due to the fact that the expander that we obtain is a device constructed completely from cobalt-chrome, stiffer than all the previous expanders described to date.

Furthermore, some studies have seen that mandibular displacement is a frequent phenomenon in the case of unilateral crossbite [29,30]; however, with the use of HIRME, the partial occlusal coverage provided by the rigid cobalt-chrome structure has the advantage of separating the occlusion, giving the muscles the opportunity to abandon the habitual shift and the mandible to return to the centric relation during treatment, thus postponing any occlusal adjustment of the interferences left after the treatment. In particular, it has been seen that the removal of the occlusal feedback given by the occlusal elevation of the expander makes the muscles, as well as the joints, return to work in a symmetrical way, fully expressing their function, if there are no other pathologies or alterations.

In the future, the goals will be to look for a hypoallergenic material, biocompatible even for those patients who are nickel allergic, and to try to develop an even less bulky expander with the use of smaller expansion screws and, possibly, with the use of an even more rigid material that allows the development of a less extensive, and therefore less bulky, shield.

## 4. Conclusions

This article shows that a HIRME, as well as a Haas-type RME appliance anchored to the primary teeth, is an effective device for solving posterior crossbite and the maxillary transverse deficit and for providing space for the maxillary permanent lateral incisors in the mixed dentition; in fact, at this dental age, it is possible to band the second primary molars without involving the permanent dentition, and this consequently avoids the risk of damage to these teeth. Furthermore, the anterior maxillary dentition improves spontaneously toward a low level of irregularity, which may reduce the prevalence of patient requests to align the anterior maxillary dentition.

There are many advantages in terms of speed of application and patient compliance because no impression is required for the construction. The main feature of the HIRME is that it is developed with a completely digitized and individualized process for the mouths of young patients. The flow is completely digital, from the first moment of taking the impression with an optical scanner to the final solidification phase by the use of a 3D printer; the “raw” device is then finished in the laboratory and sent to the dentist, ready to be cemented directly onto the patient’s mouth. This entire process is simpler and much faster than the traditional one adopted in the daily clinical practice of many dental offices.

## Figures and Tables

**Figure 1 materials-13-02898-f001:**
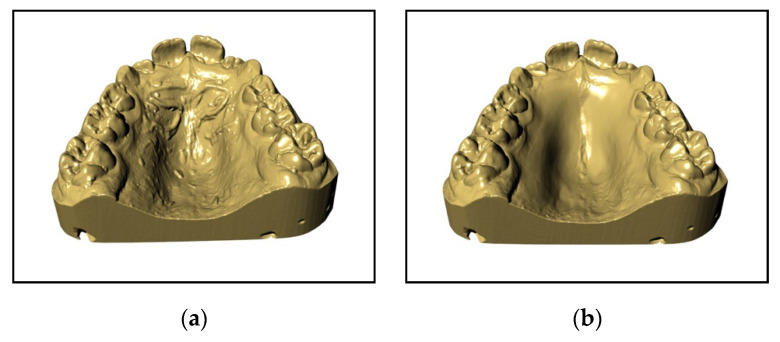
(**a**) Digital model of the upper arch. (**b**) Digital model of the upper arch after being smoothed.

**Figure 2 materials-13-02898-f002:**
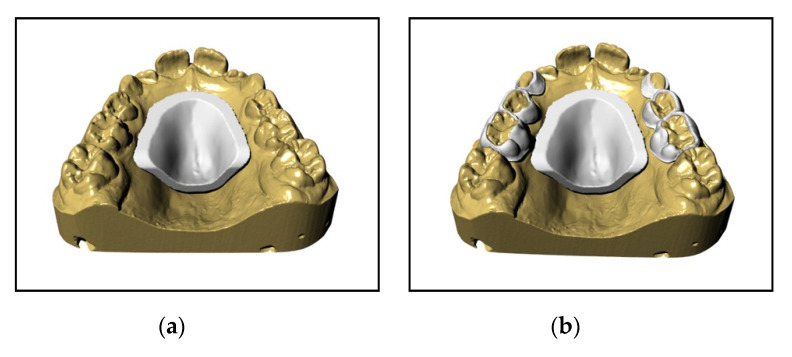
(**a**) Shield design. (**b**) Support structures built upon deciduous teeth. (**c**) Arms that connect the shield to the support structures. (**d**) Shield with support surfaces for the screw. (**e**) Final device complete with all accessories.

**Figure 3 materials-13-02898-f003:**
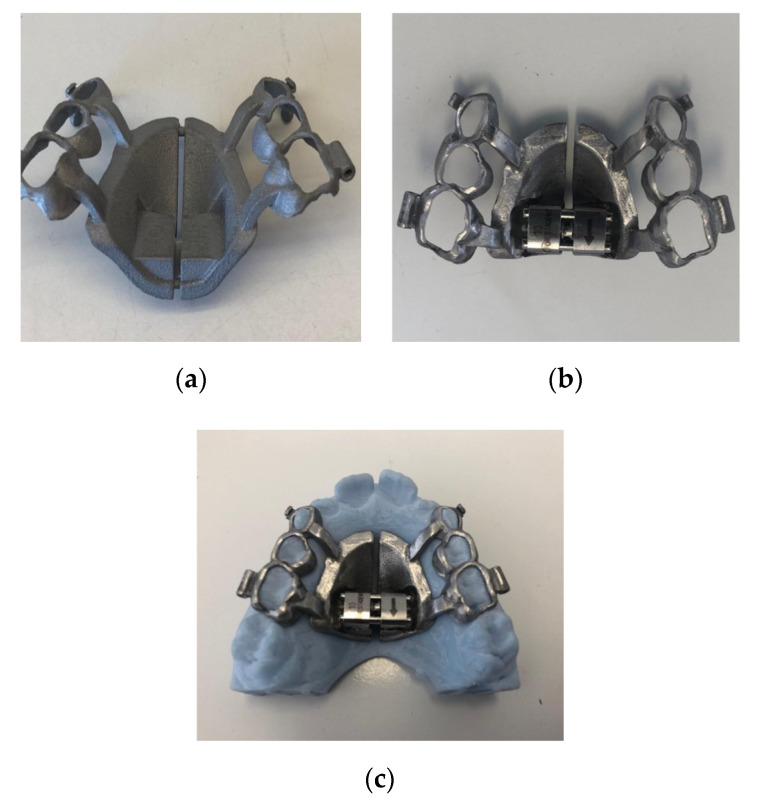
(**a**) “Rough” structure before “post printing process". (**b**) External polishing of the device. (**c**) Checking the device on the model.

**Figure 4 materials-13-02898-f004:**
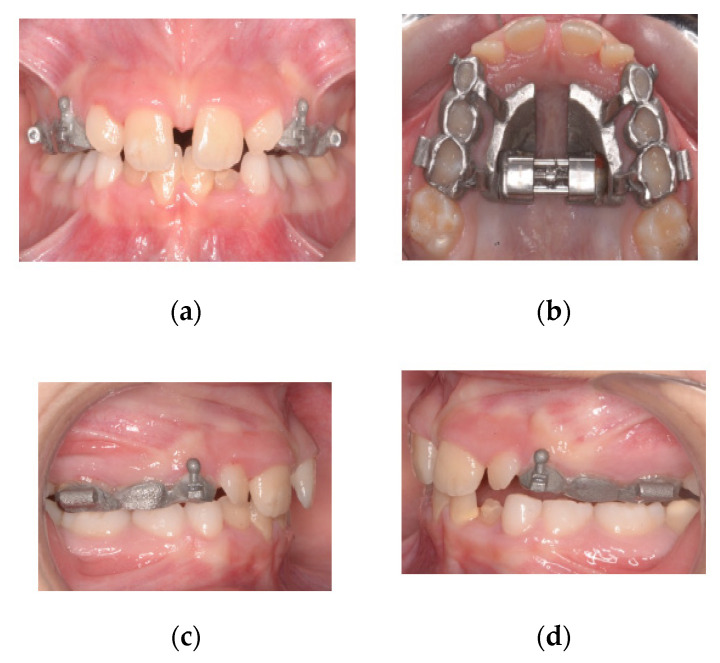
(**a**) Frontal photo of the patient after activating the expander. (**b**) Screw blocked with metal binding. (**c**) Right side photo. (**d**) Left side photo.

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
