# Peer review of "A New Proposal: A Digital Flow for the Construction of a Haas-Inspired Rapid Maxillary Expander (HIRME)"

_materials, 2020, doi:10.3390/ma13132898_

Round 1

Reviewer 1 Report

The author here proposes a digital flow for the construction of 3 a Haas Inspired Rapid Maxillary Expander (HIRME). I would recommend for publication after revision.

1) the author states the use of metal, was any test conducted for the lifetime of the material in the environment around teeth. 

2) was any precautionary measures taken to stop rusting? 

3) What is the extend of expander which can be introduced without causing any wear and tear?

4) What the effect of stress and strain on the muscle of the patient? 

Reviewer 2 Report

I read your manuscript with interest.

There are a few many errors along the manuscript that must be corrected. This errors are related to the format of the text, figures legends, and in general, missing to to follow the guidelines as close as is expected for publication consideration.
The last sentence of the conclusion section shall be eliminated. 
This manuscript needs thorough attention to detail. 
Once this is corrected, the manuscript may considered more seriously for acceptance. 

Author Response

The manuscript was checked by native English speaker.

Some images that were unfortunately uncorrected and their legends have been replaced and correct in the "Procedure" section.

The last sentence of the "Conclusion" section has been removed and, as you can see in the manuscript, the discussions have been expanded.

Reviewer 3 Report

In the time we try to transfer our workflow to digital as much as possible this article follow this modern concept and because of that can be interesting to the readers, but it does not have high scientific value or significance of content. Because of that is hard to suggest that it should be published in scientific journal, it may be more appropriate for clinical orthodontic journal.

Reviewer 4 Report

Dear authors

I have carefully read your manuscript with great interest.

I think that it should sound very interesting for readers but this paper include clearly limitation.

Oral impression is performed by an optical scanner. it looks like without validation of scanner's precision and trueness in digital flow.

   --> Although there are many advantages in terms of speed application and patient compliance because no impression is required for the construction, validation is very important point.

So, I think that author need to explain for the precision of their procedure.

Sincerely,

Round 2

Reviewer 2 Report

Thank you for working on the suggesitons.